# The Influence of Carbon Nature on the Catalytic Performance of Ru/C in Levulinic Acid Hydrogenation with Internal Hydrogen Source

**DOI:** 10.3390/molecules25225362

**Published:** 2020-11-17

**Authors:** Marcin Jędrzejczyk, Emilia Soszka, Joanna Goscianska, Marcin Kozanecki, Jacek Grams, Agnieszka M. Ruppert

**Affiliations:** 1Institute of General and Ecological Chemistry, Faculty of Chemistry, Lodz University of Technology, Żeromskiego 116, 90-924 Łódź, Poland; e.soszka1991@gmail.com (E.S.); jacek.grams@p.lodz.pl (J.G.); 2Faculty of Chemistry, Adam Mickiewicz University in Poznań, Uniwersytetu Poznańskiego 8, 61-614 Poznań, Poland; asiagosc@amu.edu.pl; 3Department of Molecular Physics, Faculty of Chemistry, Lodz University of Technology, Żeromskiego 116, 90-924 Łódź, Poland; marcin.kozanecki@p.lodz.pl

**Keywords:** levulinic acid, formic acid, Ru/C, γ-valerolactone, biomass conversion, carbon materials

## Abstract

The influence of the nature of carbon materials used as a support for Ru/C catalysts on levulinic acid hydrogenation with formic acid as a hydrogen source toward gamma-valerolactone was investigated. It has been shown that the physicochemical properties of carbon strongly affect the catalytic activity of Ru catalysts. The relationship between the hydrogen mobility, strength of hydrogen adsorption, and catalytic performance was established. The catalyst possessing the highest number of defects, stimulating metal support interaction, exhibited the highest activity. The effect of the catalyst grain size was also studied. It was shown that the decrease in the grain size resulted in the formation of smaller Ru crystallites on the catalyst surface, which facilitates the activity.

## 1. Introduction

Biomass is a source of a wide range of chemicals, called platform molecules, which are used as intermediates in a variety of industrial processes [1,2]. One of the most important compounds directly obtained from lignocellulosic biomass is levulinic acid (LA), being a product of acid hydrolysis of carbohydrates [3]. Levulinic acid can be converted into various important derivatives, e.g., γ-valerolactone (GVL). GVL is synthesized in the hydrogenation of levulinic acid (Equation (1)) [4]. Thanks to its properties, it can be used as a green solvent, fuel additive, polymer monomer, or intermediate of further compounds such as valeric acid (VA), 1,4-pentanediol (1,4-PDO), and 2-methyltetrahydrofuran (2-MTHF) [5,6,7].
Levulinic acid + H_2_ → GVL + H_2_O     ∆*H_gas_* = −47.73 kJ·mol^−1^, ∆*H_liq_* = −74.5 kJ·mol^−1^(1)

Formic acid (FA) that is formed in the equimolar amount with levulinic acid can be used as an internal hydrogen source in the synthesis of γ-valerolactone [8,9]. The hydrogenation of LA to GVL requires the application of a suitable catalyst. The catalyst for the hydrogenation reaction must contain active centers for the selective formic acid decomposition and subsequent hydrogenation of LA [10]. The decomposition of formic acid takes place in two ways via dehydrogenation and dehydration, whereby the first reaction is preferred [11]. The products of the dehydrogenation reaction of FA are hydrogen and carbon dioxide (Equation (2)), while the dehydration reaction leads to the formation of carbon monoxide and water (Equation (3)) [12].
HCOOH → H_2_ + CO_2_     ∆*G*° = −32 kJ·mol^−^^1^, ∆*H*° = 31 kJ·mol^−1^, ∆*S*° = 216 J·mol^−1^·K^−1^(2)
HCOOH → CO + H_2_O     ∆*G*° = −12.4 kJ·mol^−1^, ∆*H*° = 29.2 kJ·mol^−1^, ∆*S*° = 139 J·mol^−1^·K^−1^(3)

It is obvious that the dehydration reaction reduces the selectivity to the desired hydrogen. Furthermore, carbon monoxide can poison metallic centers due to its strong adsorption ability on metals that may decrease their activity [13,14,15].

Ruthenium-based systems are the most active heterogeneous catalysts for the hydrogenation of levulinic acid with an external hydrogen source for reactions conducted in the water phase [16,17,18]. However, the activity of ruthenium catalysts in the hydrogenation of LA to GVL strongly depends on the nature of the support [19,20,21,22]. Due to the high acidity of supports and optimal size of metal crystallites, ruthenium catalysts based on H-β and H-USY zeolites proved to be efficient in the hydrogenation reaction of LA to GVL [23]. On the other hand, Luo et al. have shown that although the high acidity of H-β and H-ZSM-5 zeolite materials accelerates the hydrogenation reaction of LA to GVL, it also pushes the reaction forward toward pentanoic acid [24].

It was found that Ru dispersion depends on the crystalline phase of TiO_2_. The rutile phase of titanium oxide facilitates the formation of small metal crystallites, whereas anatase favors the formation of large aggregates [25]. The optimum size of Ru crystallites for Ru/TiO_2_ giving the highest activity in the LA hydrogenation was also established [26]. However, the strong metal–support interaction observed in the case of Ru-TiO_2_ catalyst leads to its partial deactivation in subsequent reaction cycles due to the partial covering of metal crystallites with a support that is induced by the reaction conditions [27]. A comparison of ruthenium catalysts based on Al_2_O_3_, SiO_2_, or active carbon in the hydrogenation reaction of levulinic esters showed that the metal strongly interacts with the silica and alumina, which reduces the yield to GVL. In turn, the weaker interaction of the metal with the carbon surface increases the number of active centers that are more active in the hydrogenation of LA [28].

Ru/C, which is typically considered as a benchmark industrial catalyst, was used in the LA hydrogenation mainly for the optimization of process conditions [29], establishment of reaction kinetics [30,31], or understanding of the reaction mechanism [32]. However, the potential of carbon modification was often hindered. Despite this, carbon materials are known for their high stability and large surface area [33]. However, there are very interesting studies showing that modification of the carbon nature can have a strong influence on LA hydrogenation. The application of graphene-supported catalyst leads to the formation of electron-enriched ruthenium crystallites with enhanced activity in LA hydrogenation [34]. In addition, graphene used as a support prevents the migration and aggregation of Ru crystallites, which allows maintaining high activity and selectivity in LA conversion to GVL [19]. Moreover, Wei et al. showed that the interaction of ruthenium with nitrogen-modified ordered mesoporous carbon (OMC) materials leads to higher stability and better metal dispersion in LA hydrogenation [35]. On the other hand, Gallegos-Suarez et al. [36,37] found that the modification of active carbon used as a support can lead to a decrease in ruthenium dispersion due to the higher content of oxygen functional groups on the surface, which can result in an increase of the activity in the glycerol hydrogenolysis reaction [38].

As it was shown, the activity of the Ru/C catalyst depends strongly on the nature of the carbon materials. However, the information concerning the influence of the structural and chemical properties of carbon supports on the activity of ruthenium catalysts in the LA hydrogenation is limited. Taking this into account, we focused on the investigation of the influence of physicochemical properties of carbon materials on the selectivity and activity of Ru/C catalysts in the hydrogenation of levulinic acid with formic acid used as a hydrogen source. We paid special attention to the relationship between the metal–support interaction, hydrogen mobility (related with the surface structure of carbon materials), and catalytic performance.

## 2. Results

### 2.1. Textural Properties of Carbon Supports

Commercially available carbon materials Norit, AG, CWZ, and AC with a grain size below 0.10 mm were used as supports. Moreover, additional fractions of AC material with a grain size in the range of 0.25–0.50 mm and 0.75–1.00 mm were investigated. The textural properties of carbon materials are shown in Table 1 (additionally, nitrogen adsorption–desorption isotherms are presented in Appendix A). All supports are characterized by a mesoporous structure with specific surface area in the range of 649–973 m^2^/g, pore volume 0.090–0.652 cm^3^/g, and an average pore diameter of 3.4–5.7 nm. AG is characterized by the highest specific surface area 973 m^2^/g and the highest pore volume 0.652 cm^3^/g among studied carbon materials. In contrast, Norit reveals the lowest specific surface area 649 m^2^/g and the pore volume 0.421 cm^3^/g and the highest average diameter of pores 5.7 nm. In turn, CWZ exhibits a smaller specific surface area of 789 m^2^/g and total pore volume of 0.211 cm^3^/g than AG but a larger average pore diameter: 4.7 nm.

In the case of different fractions of AC, a decrease in the specific surface area from 882 to 691 m^2^/g was observed together with the decrease of the fraction range. This is accompanied by the decrease of the total pore volume.

### 2.2. Scanning Electron Microscopy

In the case of all carbon materials, the SEM images (Figure 1) showed a uniform surface with fine grain size, only for AC1, larger grains were visible. In addition, higher magnification images showed that some of the grains of the CWZ sample have a pillar shape structure, whereas other materials possessed irregular grains. The SEM images of AC-based samples demonstrates the decrease in grain size due to gridding. In addition, SEM images of the AC3 sample at higher magnification reveal its expanded porous structure. The structure of AC2, as a result of grinding, is partially destroyed. Furthermore, the AC1 sample shows only small grains. The observed partial degradation of the porous structure of the AC is consistent with the results of the specific surface area of carbon materials, which showed its decrease with a simultaneous reduction in the size of the support grains.

### 2.3. Raman Spectroscopy

Raman spectra of supports and catalysts were collected to determine the changes in the structure of carbon materials caused by metal impregnation (Figure 2 and Figure 3).

All spectra show two characteristic bands at around 1355 and 1590 cm^−1^. The band at higher frequency called the G band is attributed to plane vibrations of sp^2^-bonded carbon atoms of the ordered structure with a high degree of symmetry. In turn, the band at lower frequency described as the D band is characteristic for the defected and disordered structure of carbon materials [39,40,41]. The *I_D_/I_G_* ratio was calculated for each sample, and the results are included in Figure 2 and Figure 3 [42]. Differences in the contribution of disordered and ordered phases were observed in the case of the structure of bare carbon materials, and final catalysts were evidenced. The *I_D_/I_G_* ratio for supports decreases in the following order: AC1 > CWZ ≈ Norit > AG (from 1.89 for AC1 to 1.41 for AG, respectively). It means that the structure of the AC1 has the largest number of defects in contrast to the AG sample in which the contribution of disordered phase in the structure is the lowest. An impregnation of carbon materials by ruthenium slightly changes the content of both phases in the structure of catalysts. The *I_D_/I_G_* ratio of the Ru/AC1, Ru/CWZ, and Ru/Norit catalyst is lower in comparison to bare supports. This suggests that the introduction of Ru increases the contribution of the ordered phase in the structure of carbon materials [43]. On the contrary, in the case of Ru/AG, there are no significant changes in the *I_D_/I_G_* ratio of the support and catalyst.

### 2.4. Phase Analysis of Catalysts

XRD analysis was performed to characterize the phase composition of ruthenium catalysts supported on different carbon materials (Figure 4).

The diffractograms show broad characteristic signals at 25° and 44° assigned to the (002) and (101) planes of the amorphous structure of the carbon (ICDD:13-0148), while signals at 21°, 26°, and 36° are ascribed to the (100), (011), and (110) diffraction of the silica (ICDD:01-070-7345), which is a typical impurity of carbon materials [44,45]. It is worth noting that the signal at 26° can be assigned both to the silica and (002) plane of the graphite (ICDD:00-041-1487). The signals that originate from silica or graphite are not present in the case of Ru/CWZ. In addition, the broad signals at 38° and 44° are assigned to the (100) and (101) diffraction of metallic ruthenium (ICDD: 00-006-0663). The shift of the signal in the case of the Ru/AG sample might be related with the change of the size of the ruthenium unit cell [46]. 

### 2.5. Temperature-Programmed Desorption of NH_3_ and CO_2_

Temperature-programmed desorption of ammonia and carbon dioxide were performed to assess the acid–base properties of supports and catalysts (Table 2). 

The oxygen-containing functional groups such as carboxyl, anhydride, lactone, lactol, or hydroxyl possessing acidic character and pyrone-like and chromene groups having basic character were identified on the surface of carbon materials [47]. Acid centers predominate on the surface of supports and catalysts. The acidity of CWZ and AC1 (118 and 154 µmol/g, respectively) is much higher than the acidity of Norit and AG (32 and 58 µmol/g, respectively). The same trend has been observed for the basicity of studied carbon materials. Ruthenium impregnation increases both the acidity and basicity of catalysts compared to bare supports. This is related with the type of the used preparation method. Namely, the metal was introduced from acidic aqueous solution of ruthenium chloride. This treatment resulted in the shift of the acid–base balance of the surface toward the acid side [48]. Chen et al. showed that treatment with non-oxidizing acid increases the number of weak acid groups on the surface of activated carbon due to transformation of double-bonded oxygen groups to single-bonded oxygen groups and an increase in the oxygen content. The latter phenomenon is a result of chemisorption of the water molecule by delocalized π electrons in the carbon planes [49]. In addition, the higher acidity and basicity of catalyst compared to bare support can be related to the presence of chloride ions on its surface. Perez-Cadenas et al. reported that due to their electron properties, chloride ions on the surface of carbon materials weaken Brönsted acidity while increasing Lewis acidity as a result of the resonance effect [50].

### 2.6. Chemisorption of CO

Dispersion and the average size of ruthenium crystallites on the surface of the catalysts were determined based on the amount of adsorbed carbon monoxide, using the formula given by Newman et al. [51]. The results are presented in Table 3. The metal dispersion on the surface of CWZ, AG, Norit, and AC1 is comparable and in the range of 14.9–16.2%, as well as the average size of ruthenium crystallites, which is in the range of 2.7–2.8 nm. The results are surprising considering the differences in the specific surface area and acidity of the surface of the AC1, Norit, CWZ, and AG samples and especially that the oxygenated groups of carbon materials can play a significant role in the dispersion of ruthenium [37,52]. The surface groups improve the hydrophilicity of the carbon surface, which may increase the interaction of the metal precursor with the support improving the dispersion of metal [53]. However, the specific surface area or acidity of AC1, CWZ, Norit, or AG do not affect noticeably the dispersion and size of ruthenium crystallites. In contrast, chemisorption data for Ru/AC3, Ru/AC2, and Ru/AC1 catalysts showed that the metal dispersion increases from 5.4% to 15.1% with a decrease in the grain size and specific surface area of the AC support.

### 2.7. Temperature-Programmed Reduction

A temperature-programmed reduction of hydrogen was performed to assess the reducibility of the catalysts (Figure 5); for comparison, the temperature-programmed reduction (TPR) of bare supports is also shown (Appendix A). The TPR profiles of catalysts show two ranges of hydrogen consumption at low temperature (from 100 to 300 °C) and at high temperature (above 400 °C) regardless of the type of support used. Low temperature effect, in the range of 100–300 °C, corresponds to the reduction of electron-deficient ruthenium species to metallic ruthenium. It is worth noting that in this case, the reduction of ruthenium species proceeds in two stages, as indicated by two visible maxima in TPR profiles slightly below and above 200 °C. It has been shown that two kinds of ruthenium species such as RuCl_3_ and RuO_2_ were evidenced in the case of catalyst obtained by the wet impregnation method with RuCl_3_ as a metal precursor [51]. Moreover, the reduction of ruthenium oxide occurs at a higher temperature compared to ruthenium chloride [54,55]. Wang et al. showed that a decreasing amount of chlorine in the ruthenium catalyst promotes the formation of metal oxide and at the same time shifts the reduction toward higher temperature [56]. Thus, the hydrogen consumption peak located below 200 °C is related to the reduction of RuCl_3_, while the peak observed above 200 °C is attributed to a reduction of ruthenium oxide. The amount of hydrogen consumed in each step depends on the type of support. In case of ruthenium supported on AG, Norit, and CWZ, the peaks below 200 °C have a higher intensity than the peaks above 200 °C. An inverse relationship can be observed in the profile of AC1 catalysts where the peaks above 200 °C are more intense than those below 200 °C. This implies that the surface of AC carbon promotes the formation of ruthenium oxide during drying, while the Norit, CWZ, and AG5 surfaces rather stabilize adsorbed RuCl_x_ during catalyst preparation. This can be a result of differences in the metal support interaction. Raman spectra revealed that the structure of carbon materials characterizes both ordered and disordered phases. The AC1 material has the largest proportion of disordered phase in the structure among the tested materials. These defective sites are excellent centers for anchoring ruthenium crystallites [57]. In addition, the TPR profiles of the Ru/AC1 catalysts show the highest content of ruthenium oxide. Thus, the surface defects of the AC1 material probably are responsible for the strong interaction of the support with the metal precursor, which promotes the oxidation of ruthenium chloride to the oxide during drying. In addition, for AC-based samples, a peak shift above 200 °C toward lower temperatures can be observed as the support grain size increases. This phenomenon is associated with smaller dispersion and larger metal crystallites that can be easily reduced, since unsupported RuO_2_ crystallites with larger diameter are reduced at about 100 °C [58].

The hydrogen consumption at high temperature above 400 °C is attributed to the reduction of support near ruthenium crystallites. It can be also observed in the TPR-H_2_ profiles of bare supports, but it started at higher temperatures than in the case of catalysts. Metal on carbon materials catalyzes the gasification of the carbon materials probably due to the higher hydrogen activity on the surface of the metal crystallites [59].

### 2.8. Temperature-Programmed Desorption of H_2_

The temperature-programmed hydrogen desorption was performed in order to check the strength of metal interactions with hydrogen. The TPD-H_2_ curves collected for the catalysts are shown in Figure 6. The desorption of hydrogen from the surface of all catalysts begins at 300 °C; however, the maximum desorption peak depends on the type of support. In the case of the Ru/AC1 catalyst, the maximum H_2_ desorption was observed at 419 °C, while for catalysts based on CWZ and Norit supports, the maximum desorption peak is visible at 460 °C and 441 °C, respectively. In turn, for the Ru/AG system, the maximum hydrogen desorption was recorded at 490 °C. Li et al. showed that the desorption of hydrogen from the surface of ruthenium catalysts can be related to the presence of chemisorbed hydrogen and a spillover hydrogen [60]. Decrease of the temperature of the hydrogen desorption indicates a weaker adsorption of H_2_ and its greater mobility on the catalyst surface. Hydrogen desorption from the surface of the Ru/AC1 catalyst takes place at lower temperature than for other catalysts. In general, based on TPD-H_2_, the mobility of hydrogen on the surface of catalysts can be ordered in the following sequence: Ru/AC1 > Ru/Norit > Ru/CWZ > Ru/AG.

### 2.9. Catalytic Activity

Formic acid decomposition and subsequent hydrogenation of LA (FALA reaction) with Ru supported on different carbon materials was performed. In order to understand the catalytic performance, the independent reactions were carried out as well. The results are presented in Table 4 and Table 5.

In the formic acid decomposition, the highest activity was obtained for Ru/AC1 catalyst (97% conversion). The Ru/Norit, Ru/CWZ, and Ru/AG catalysts showed 84%, 80%, and 77% conversion of formic acid, respectively. All catalysts showed high selectivity to hydrogen. The selectivity to carbon monoxide for the latter catalysts did not exceed 5%. In turn, Ru/AC1 catalyst had a bit higher selectivity to carbon monoxide 7%. In the case of Ru/AC2 and Ru/AC3 catalysts, the conversion of formic acid was lower and the selectivity to carbon monoxide of both catalysts was slightly higher than for the Ru/AC1 catalyst.

In the case of hydrogenation of levulinic acid to gamma-valerolactone with an external hydrogen source, the highest substrate conversion (95%) and yield to product (78%) were noted for the Ru/AC1 catalyst. The conversion of levulinic acid with Ru/Norit Ru/CWZ and Ru/AG catalysts was similar and reached 80%. In addition, these catalysts showed similar yield to the reaction product (in the range 64–71%). The most active catalyst shows also high stability under the reaction conditions (Appendix A).

The results of the simultaneous decomposition of formic acid and hydrogenation of levulinic acid are presented in Table 5. It is worth noting that all catalysts in the FALA reaction showed 100% conversion of formic acid. The differences in catalyst activity are only visible in LA conversion and in the yield to GVL. Ru/Norit and Ru/CWZ catalysts showed 38% and 35% yield for GVL, respectively. In turn, the lowest activity of only 18% GVL in the FALA reaction was revealed by the Ru/AG. The highest yield to GVL of 59% was obtained for the Ru/AC1 catalyst, whereas catalyst supported on other AC fractions showed lower activity.

## 3. Discussion

Levulinic acid hydrogenation with formic acid used as a hydrogen source requires that the catalyst is both selective toward hydrogen production and active in the hydrogenation process. Activity tests with Ru/C materials show that there are several factors responsible for catalytic performance in the studied reaction with metal–support interaction and the strength of hydrogen adsorption being the most important. Those factors are directly related with the nature of the carbon material.

The influence of the size of Ru crystallites although known as the important factor [26] could be excluded in our case due to the uniform dispersion of Ru on the supports among tested catalysts.

There are two factors contributing to the strength of metal–support interaction: the presence of defects on the surface of carbon material and the type of Ru species. The most active Ru/AC1 showed the highest number of defects, which stimulate the strong interaction of metal with the support [57]. Moreover, in the case of this catalyst, the highest contribution of RuO_x_, which is less prone to be reduced, was identified. On the other hand, lower number of defects in the case of Norit, CWZ, and AG materials rather stabilize ruthenium chloride on their surface.

The strength of hydrogen adsorption is considered equally important. The hydrogen desorption temperature revealed from TPD studies correlates with the results of catalyst activity. The lowest hydrogen desorption temperature is observed for Ru/AC1, so the highest mobility of hydrogen enhances the activity of this catalyst to the highest extend, while Ru/AG exhibiting the highest hydrogen desorption temperature shows the lowest activity in the reactions.

In the case of the decomposition of formic acid, a strong adsorption of hydrogen can poison the catalyst [61]. Higher hydrogen mobility on the catalyst surface reduces certainly the effect of catalyst poisoning and simultaneously increases its activity in the FA decomposition reaction. It is worth noting that both the decomposition of FA and LA hydrogenation occur at the same centers [32]. Thus, the higher mobility of the hydrogen molecules on the surface of the catalyst, due to the strong interaction of ruthenium with the AC surface, limits the poisoning of active sites of catalysts in the formic acid decomposition reaction and at the same time results in higher activity in the FALA reaction.

In addition, considering Ru/AC1, Ru/AC2, and Ru/AC3 systems, there is a close relationship between the activity and the grain size of the support. Ru supported on the grains below 0.1 mm (AC1) showed higher activity in all investigated reactions than catalysts with larger grain sizes (Ru/AC2 and Ru/AC3). Chemisorption studies have shown that a smaller grain size of the support increases the dispersion of ruthenium. Moreover, a smaller grain size decreases the possible diffusion limitations and increases the availability of active centers for substrates.

## 4. Materials and Methods

### 4.1. Catalysts Preparation

Four types of commercially available carbon materials were used as catalyst supports. The AC was supplied by Windsor Laboratories, Ltd., Slough-Berkshire, UK. The Norit (grain size < 0.1 mm) and CWZ (grain size < 0.1 mm) were purchased by ChemPur, Piekary Śląskie, Poland. AG (grain size < 0.1 mm) was delivered by Gryfskand, Gryfino, Poland. In order to determine the effect of grain size of the support, starting carbon AC has been crushed in mortar and sieved to obtain the following fractions < 0.1; 0.25–0.5; 0.75–1.00 mm (AC1, AC2, and AC3, respectively). Catalysts were prepared by the wet impregnation method using water solution of RuCl_3_ (100% pure, Merck, Darmstadt, Germany) to obtain 5 wt % of metal on the support surface. After 24 h impregnation, an excess of solution was evaporated, and catalyst was dried at 120 °C for 2 h and reduced at 500 °C for 1 h under hydrogen flow before the activity tests.

### 4.2. Catalytic Activity Tests

The catalytic activity was tested in the levulinic acid hydrogenation with external and internal source of hydrogen. Reactions were carried out in a stainless-steel autoclave (Berghof, Eningen, Germany) equipped with teflon insert, allowing a reaction volume of 45 mL. In the case of the application of an external source of hydrogen, the reaction was performed under the pressure of 10 bar of hydrogen. In the second case, formic acid was used as an internal hydrogen source. In typical levulinic acid hydrogenation, 1 g of LA, 0.3 g of catalyst, and 30 mL of distilled water were used. In turn, in a combined levulinic acid hydrogenation with formic acid as a source of hydrogen (FALA reaction) 1 g of LA, 0.4 mL of FA, 0.6 g of catalyst, and 30 mL of distilled water were used. The temperature of reaction was maintained at 190 °C for 2 h for FALA reaction or 1 h for levulinic acid hydrogenation. After the reaction, the reactor was cooled down to room temperature, the remaining pressure was released, and the mixture was centrifuged to separate the catalyst from the solution. The liquid products were analyzed by high-performance liquid chromatograph (Agilent Technologies 1260 Infinity, Perlan Technologies, Santa Clara, CA, USA) equipped with refractive index detector and Rezex ROA column, using 0.0025 mol∙dm^−3^ H_2_SO_4_ as an eluent.

Formic acid decomposition was carried out in a homemade flow reactor at atmospheric pressure. Prior to the reaction, the catalyst (0.03 g) was reduced under hydrogen flow at 500 °C for 1 h. Next, the reactor was cooled down to reaction temperature under Ar flow. Formic acid was continuously introduced into the catalyst bed together with the flow of Argon. The gas line was heated to avoid the condensation of gas samples. The reaction products were analyzed on a gas chromatograph Hewlett Packard 5890 (Palo Alto, CA, USA) equipped with a TCD (Thermal Conductivity Detector) detector and a 6 m Porapak Q column.

### 4.3. Materials Characterization

The specific surface area measurements of carbon materials were carried out on ASAP 2010 Micromeritics (Micromeritics Instrument Corporation, Norcros, GA, USA) with nitrogen as the adsorbate. Before nitrogen adsorption, the sample was outgassed at 200 °C for 3 h to remove water and impurities from its surface. The specific surface area was determined based on the BET (Brunauer–Emmett–Teller) adsorption isotherm and the pore distribution was calculated based on the BJH (Barrett-Joyner-Halenda) nitrogen desorption isotherm.

Temperature-Programmed Reduction (TPR) was performed on an AMI1 system from Altamira Instruments (Pittsburgh, PA, USA) equipped with a thermal conductivity detector. The reducibility of carbon supports and catalyst was examined with the use of the mixture of 5 vol.% H_2_ and 95 vol.% Ar at a space velocity of 3.1 × 10^−9^ g s^−1^ cm^−3^ and a linear temperature ramp of 10 °C min^−1^.

CO chemisorption studies were carried out with the use of a homemade PEAK-4 apparatus [62]. Dried catalyst was placed in a quartz tube reactor and was in situ reduced at 650 °C for 1 h in H_2_ stream with a flow rate of 40 cm^3^ min^−1^. Then, the flow of H_2_ was switched to argon, and the reactor was cooled to room temperature. Carbon monoxide was introduced into the reactor by pulses using a six-way valve. An infrared gas analyzer (Fuji type ZRJ-4, Fuji Electric, Tokyo, Japan) was used to monitor changes in CO concentration.

The temperature-programmed desorption of NH_3_ and CO_2_ (TPD-NH_3_ and TPD-CO_2_) was used to study the acid–base properties of supports and catalysts. In the typical TPD of ammonia or carbon dioxide protocol, a dried sample was placed in a quartz flow reactor and reduced in situ at 500 °C under hydrogen flow for 1 h. After the change of hydrogen to helium, the temperature was decreased to 100 °C and adsorption of NH_3_ or CO_2_ was carried out for 15 min. Next, physically adsorbed probe molecules were removed from the sample surface by treating the sample with He for 15 min and subsequently cooled down to room temperature. The TPD experiments were carried out from room temperature to 500 °C using a linear temperature ramp (25 °C min^−1^) and TCD detector.

The temperature-programmed desorption of hydrogen (TPD-H_2_) was carried out according to the following procedure. The dried catalyst was placed in a homemade oven and then reduced under hydrogen atmosphere to 500 °C. After switching to argon, the catalyst was maintained for another hour at this temperature. Then, the catalyst was cooled to 100 °C under argon flow. Measurements were recorded from 100 to 550 °C with a constant rate of temperature growth (10 °C/min) in the argon flow, and the amount of desorbed hydrogen was analyzed using a TCD detector.

X-ray diffraction (XRD) measurements were collected using a PANalytical X’Pert Pro MPD diffractometer (Malvern PANalytical, Malvern, UK). The X-ray source was a copper long fine focus X-ray diffraction tube operating at 40 kV and 30 mA. Data were collected in the 5–90° 2θ range with 0.0167° step. Crystalline phases were identified by references to the ICDD PDF-2 (version 2004) database. All calculations were performed with X’Pert High Score Plus computer program (Malvern Panalytical Ltd., Malvern, UK).

Raman spectra were collected with use of a dispersive T64000 triple-grating Raman spectrometer (HORIBA Jobin-Yvon, Longjumeau Cedex, France) equipped with confocal microscope BX-40 (Olympus, Tokyo, Japan). As an excitation source, an Ar-ion laser line 514.5 nm was selected. The laser power measured on the sample surface did not exceed 2 mW to protect the sample against local overheating and eventual degradation. The acquisition time was 120 s, and the spectral resolution was below 1 cm^−1^. The spectra of the AC2 and AC3 samples and the corresponding catalysts were not collected due to strong elastic scattering on the sample grain masking the Raman signal.

The morphology of carbon supports and ruthenium catalysts was assessed with the use of scanning electron microscope (SEM) FEI Quanta 250 FEG equipped with EDS (Energy Dispersive Spectrometer) system (Hillsboro, OR, USA).

The metal content of the Ru/C catalysts was measured by the atomic absorption spectroscopy using SOLAAR M6 Unicam atomic absorption spectrometer. The weight percentage of ruthenium in the catalysts is 5 ± 0.5%.

## 5. Conclusions

Ru/C catalysts showed different activities in levulinic acid hydrogenation with formic acid used as a hydrogen source depending on the type of carbon support. In our work, we showed that the carbon nature strongly modifies the metal–support interaction and the strength of the hydrogen adsorption on the metal surface. In consequence, those factors have a direct effect on the activity of the investigated materials. The catalyst possessing the highest number of defects, stimulating metal–support interaction, exhibited the highest activity. Additionally, we found that there is a direct relationship between the strength of the hydrogen adsorption and the catalytic performance, as higher hydrogen mobility (low hydrogen adsorption strength) enhances the activity of the studied catalysts.

## Figures and Tables

**Figure 1 molecules-25-05362-f001:**
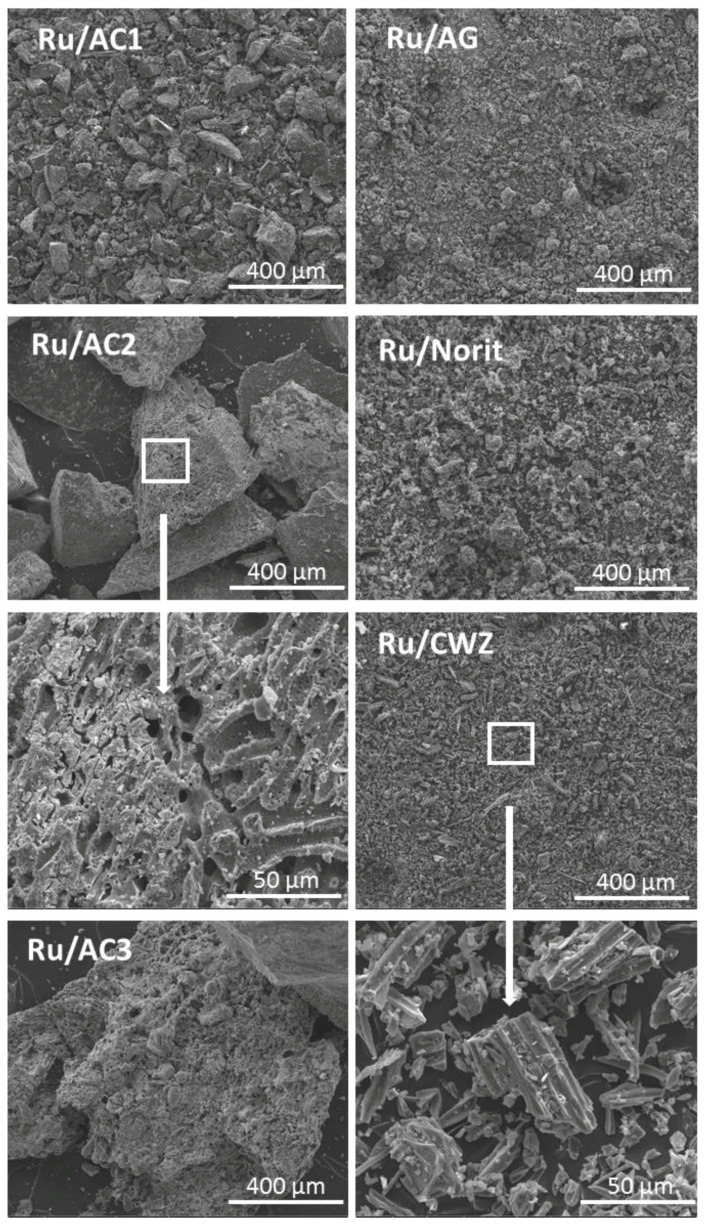
SEM images of ruthenium catalysts.

**Figure 2 molecules-25-05362-f002:**
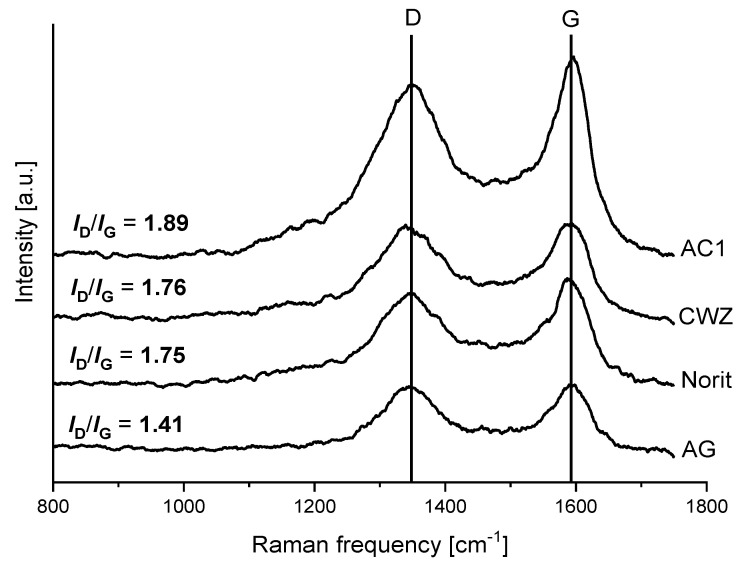
Raman spectra of carbon materials.

**Figure 3 molecules-25-05362-f003:**
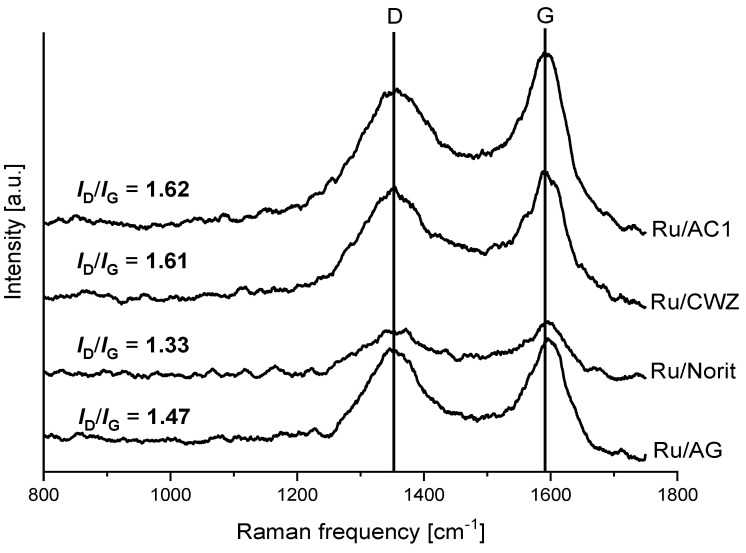
Raman spectra of ruthenium catalysts.

**Figure 4 molecules-25-05362-f004:**
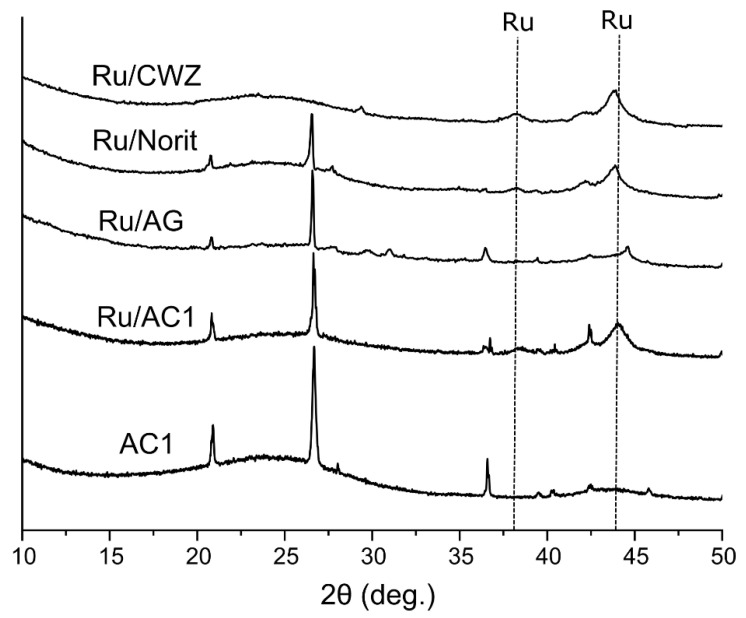
XRD pattern of AC1 support and ruthenium catalysts.

**Figure 5 molecules-25-05362-f005:**
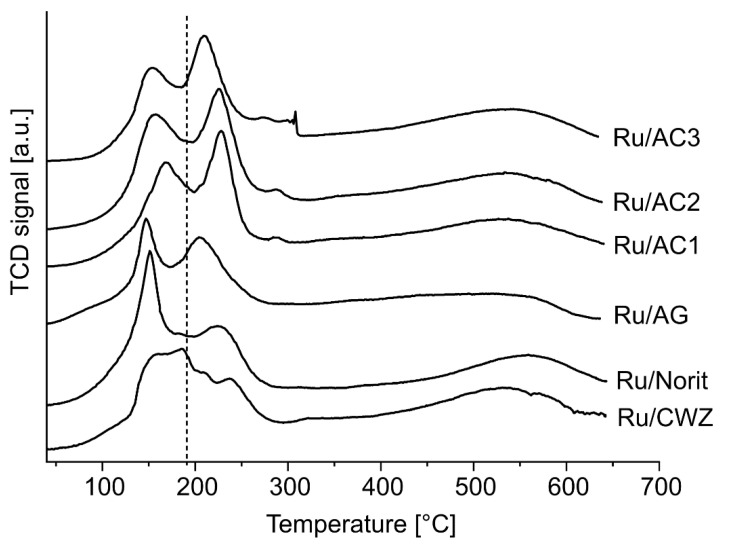
Temperature-programmed reduction (TPR-H_2_) profiles of ruthenium catalysts.

**Figure 6 molecules-25-05362-f006:**
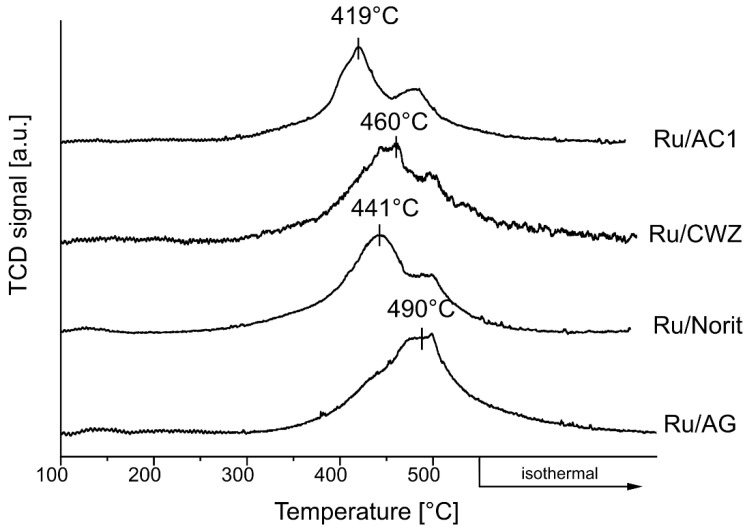
TPD-H_2_ profiles of ruthenium catalysts.

**Table 1 molecules-25-05362-t001:** Textural properties of carbon materials.

Carbon Support	Grain Size (mm)	Surface Area (m^2^ g^−1^)	Total Pore Volume (cm^3^ g^−1^)	Average Pore Diameter (nm)
Norit	<0.10	649	0.421	5.7
CWZ	<0.10	789	0.211	4.7
AG	<0.10	973	0.652	4.6
AC1	<0.10	691	0.092	3.6
AC2	0.25–0.50	724	0.090	3.4
AC3	0.75–1.00	882	0.116	3.4

**Table 2 molecules-25-05362-t002:** Acid–base properties of selected carbon materials and ruthenium catalysts.

Carbon Support	Acidity (µmol/g)	Basicity (µmol/g)	Catalyst	Acidity (µmol/g)	Basicity (µmol/g)
Norit	32	18	Ru/Norit	329	113
CWZ	154	86	Ru/CWZ	513	226
AG	58	28	Ru/AG	419	117
AC1	118	60	Ru/AC1	525	153

**Table 3 molecules-25-05362-t003:** Carbon monoxide chemisorption data, dispersion, and particle size of metal.

Catalyst	Volume of CO Adsorbed (cm^3^ g^−1^)	Dispersion of Ru from CO Chemisorption (%)	Particle Size of Ru from CO Chemisorption (nm)
Ru/Norit	1.797	16.2	2.6
Ru/CWZ	1.705	15.4	2.7
Ru/AG	1.651	14.9	2.8
Ru/AC1	1.663	15.1	2.8
Ru/AC2	0.702	6.4	6.7
Ru/AC3	0.595	5.4	7.9

**Table 4 molecules-25-05362-t004:** Activity of ruthenium catalysts in formic acid decomposition and levulinic acid hydrogenation with external hydrogen source.

Catalyst	FA Decomposition					LA Hydrogenation	
	FA Conversion (%)	Gaseous Products (% vol)				LA Conversion (%)	GVL Yield (%)
		H_2_	CO	CH_4_	CO_2_		
Ru/Norit	84	47	3	0	50	82	71
Ru/CWZ	80	47	5	0	48	80	64
Ru/AG	77	49	1	0	50	81	70
Ru/AC1	97	47	7	0	46	95	78
Ru/AC2	89	46	8	0	46	76	63
Ru/AC3	74	46	9	0	45	75	63

Reaction conditions: FA decomposition: 190 °C; 1 h; 0.03 g of catalyst; LA hydrogenation: 190 °C; 30 mL H_2_O; 1 h; 10 bar H_2;_ 0.3 g of catalyst; 1 g of LA.

**Table 5 molecules-25-05362-t005:** Activity of ruthenium catalysts in simultaneous formic acid decomposition and levulinic acid hydrogenation (FALA).

Catalyst	FALA Reaction		
	FA Conversion (%)	LA Conversion (%)	GVL Yield (%)
Ru/Norit	100	58	38
Ru/CWZ	100	58	35
Ru/AG	100	38	18
Ru/AC1	100	75	59
Ru/AC2	100	46	25
Ru/AC3	100	48	27

Reaction conditions: 190 °C; 30 mL H_2_O; 2 h; 0.4 mL of FA; 0.6 g of catalyst; 1 g of LA.

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
