# Peer review of "The Influence of Carbon Nature on the Catalytic Performance of Ru/C in Levulinic Acid Hydrogenation with Internal Hydrogen Source"

_molecules, 2020, doi:10.3390/molecules25225362_

Round 1

Reviewer 1 Report

This manuscript reports, “The influence of carbon nature on the catalytic performance of Ru/C in levulinic acid hydrogenation with an internal hydrogen source," without a doubt, a hot topic to address in the field of heterogeneous catalysis. The work is well structured and contains important results; however, it fails in originality, quality of scientific content, and contribution to existing knowledge. For instance, the same authors published (Green Chem., 18 (2016)2014) a paper with the title “Ru catalysts for levulinic acid hydrogenation with formic acid as a hydrogen source," in which they compared the catalytic activity of Ru/C, Pd/C, and Pt/C. It is suggested to make several changes before accepting its publication with major revision:

  • The authors should update the state-of-the-art of the levulinic acid hydrogenation; see, for instance, Chem Asian J., 15 (2020).
  • The authors are recommended to report the actual amount of Ru in each catalyst.
  • The acidity and basicity reported in Table 2 are two significant properties in supported metallic catalysts; however, what relationship do they have with the activity and stability of the metallic particles? Would it not be more useful to have the type of acidity (Lewis, Bronsted) to carry out a more precise correlation between properties of the solid with the catalytic properties?
  • Could the author's report (from their TPR results) the Ru precursor's reduction degree to metallic Ru? Why did oxidation not take place after the impregnation of Ru precursor and its subsequent reduction with H2?
  • The Authors did not present their CO chemisorption results, which could also verify the metal-support interaction degree.
  • Would it be possible to report the TOF values ​​in Tables 4 and 5?
  • Coke deposition and Ru leaching were they analyzed?

Reviewer 2 Report

In the abstract

  1. The phrase “It has been shown that physicochemical properties of carbon strongly affect the catalytic activity of Ru catalysts”. “One of the most crucial factors is the metal support interaction related with the presence of defects in the structure of carbon materials.”

             These facts are widely known in heterogeneous catalysis, so I believe that the Authors should describe the novelty of their research.

In the introduction

  1. The authors should include reactions, and the thermodynamic values involved.

In the results

  1. The surface area of the catalysts including BET isotherm adsorption-desorption should be included.
  2. Authors should show the index peaks of XRD analyses, for al patterns present. Also, JCPDS number should be included.
  3. By the way, In the XRD analyses, the peak related to impurities are so strong, I believe that it is present in large concentration. Do the authors believe that it would influence the reaction?
  4. Authors should explain why Ru peaks are broad, and in the case of Ru-AG is shifted towards larger Bragg angles.
  5. TPR analyses of carbons (without Ru) should show the shown.
  6. The authors should demonstrate long-term activity.
  7. Reusability of the catalysts is welcome!.
  8. Authors should compare their results with a commercial catalyst.

In general, the work is incomplete and lacks deep and extended discussion.  

Round 2

Reviewer 1 Report

I consider that the revised version substantially improves the original manuscript and answers and clears my doubts, therefore, the manuscript should be accepted for publication.

Reviewer 2 Report

The Authors must compare their results with a commercial catalyst or another present in the current literature.